# A Systematic Review of Statistical and Machine Learning Methods for Electrical Power Forecasting with Reported MAPE Score

**DOI:** 10.3390/e22121412

**Published:** 2020-12-15

**Authors:** Eliana Vivas, Héctor Allende-Cid, Rodrigo Salas

**Affiliations:** 1Escuela de Ingeniería Informática, Pontificia Universidad Católica de Valparaíso, Brasil 2950, Valparaíso, Chile; hector.allende@pucv.cl; 2Escuela de Ingeniería C. Biomédica, Universidad de Valparaíso, Chacabuco 2092-2220, Valparaíso, Chile; rodrigo.salas@uv.cl

**Keywords:** electric power, forecasting accuracy, machine learning

## Abstract

Electric power forecasting plays a substantial role in the administration and balance of current power systems. For this reason, accurate predictions of service demands are needed to develop better programming for the generation and distribution of power and to reduce the risk of vulnerabilities in the integration of an electric power system. For the purposes of the current study, a systematic literature review was applied to identify the type of model that has the highest propensity to show precision in the context of electric power forecasting. The state-of-the-art model in accurate electric power forecasting was determined from the results reported in 257 accuracy tests from five geographic regions. Two classes of forecasting models were compared: classical statistical or mathematical (MSC) and machine learning (ML) models. Furthermore, the use of hybrid models that have made significant contributions to electric power forecasting is identified, and a case of study is applied to demonstrate its good performance when compared with traditional models. Among our main findings, we conclude that forecasting errors are minimized by reducing the time horizon, that ML models that consider various sources of exogenous variability tend to have better forecast accuracy, and finally, that the accuracy of the forecasting models has significantly increased over the last five years.

## 1. Introduction

Electric power forecasting plays a substantial role in the administration and balance of current power systems. The load forecasts help to identify strategies to optimize the operating mechanisms in a determined period and thus ensure the demand even in situations adverse to the system [1]. Accompanying the rapid advances in forecasting theory [2,3] and machine learning [4,5,6], the technology in the energy forecasting research area has also developed rapidly [7]. Additionally, the popular prediction methods for the generation and demand of energy can be divided into two categories. The first category is statistical or mathematical methods, and the second category is modern statistical-learning-based methods (also known as machine learning). In addition, hybrid methods can be found that apply not only statistical tools but also other elements, such as mathematical optimization or signal processing [8,9]. Additionally, other authors [10] consider hybrid approaches that focus on a series of individual methods, such as noise reduction, seasonal adjustment and clustering, to process the data in advance, whereas combined methods use weight coefficients. With respect to the techniques implemented to forecast energy in recent years, in the international context, we can find a wide diversity; e.g., the application of kernel-based multitask learning methodologies [11], energy load forecasting methodologies based on deep neural networks as in [12,13], methodologies based on the classic time series approach as in [14,15,16], and mathematical representations as in [17,18,19]. Developing a model that achieves the highest forecasting precision in the context of electric power has been the object of study in recent years. Additionally, the determination of the appropriate input variables in load forecasting constitutes an important part of the forecasting procedure [20]. Due to the importance of the area, several review papers have appeared that present insights into current applications and future challenges and opportunities [21,22]. However, existing review papers examine the applications of a single model, e.g., an ANN [23], or cover only one energy domain, e.g., solar radiation prediction [24,25], and do not perform comparisons among specific metrics, such as MAPE, for multiple applications. Therefore, a systematic review to identify the type of model that has the highest propensity to show precision in the forecast is the main objective of this paper.

### Motivation and Scope of the Review

The number of papers published on the topic of electric power forecasting has been growing at an exponential rate throughout the last decade, as Figure 1 shows. The order of magnitude of the increase in the number of scientific publications on the subject revolves around 61.59%, between 2016–2020, with respect to 2011–2015. Generally, the studies are site-specific, and the results strongly depend on the nature of the model and the time horizon of the forecast, along with a large number of other characteristics pertaining to the data and models. This is a major limitation, which makes a generalization of the results difficult. A test of a given model over all different mentioned factors is needed to measure the average effect of the model [25]. Consequently, the contribution of this paper is to present the state-of-the-art of models in electric power systems and discuss their likely future trends, considering:(I) The models that tend to provide precision in electric power forecasts according to the literature.(II) Exogenous sources that tend to lead to accurate forecasting of electrical energy according to the literature.(III) Relationships between the times of forecasting and the accuracy of existing models.

The rest of this paper is organized as follows. In Section 2, the methodology of the research is presented. In Section 3, a description of the data set is presented. In Section 4, a performance analysis of the forecasting models is presented, and finally, the overall discussion and conclusions are presented.

## 2. Theoretical and Referential Framework

This chapter presents an analysis of the documents found in the literature during the last 15 years on the subject of electric power forecasting.

### 2.1. Selection Criteria

The number of documents published on the topic of electric power forecasting has been growing at an exponential rate throughout the last 15 years, as Figure 1 shows. We analyze in the review the documents published for electric power forecasting contained in SCOPUS, Web of Sciences, Science Direct and IEEE (Figure 2), according to the criteria shown in Figure 3 and following the steps of the PRISMA (Preferred Reporting items for Systematic Review and Meta-Analyses) methodology.

A large number of papers published between January 2005 and March 2020 were analyzed. The qualitative and quantitative synthesis of the analysis was collected from 164 documents selected based on the criteria shown in Figure 3; the documents that only forecast electric power in buildings, universities, homes, and rooms were excluded; likewise, if the time horizons are not mentioned in the Abstract, the article was also skipped in our research. Similarly, if in an article, MAPE was not used as a criterion for accuracy, it was not considered in our review. When considering the accuracy of the results reported by the selected papers in terms of the MAPE Equation (Equation 1), we can compare samples of different magnitudes, thus ensuring a common basis for intercomparison analyses.

It is important to highlight that under our filtering criteria a significant volume of valuable references may have been excluded; in this sense, our searches may not be specific. If the readers are interested exclusively in consulting documents related to forecasting under machine learning methods, then they could consult [21], or if they are interested in specific documents on solar energy, they could consult [25]; in the case of documents associated with the forecast under classical statistical techniques, there are specific documents that can be consulted, such as [26].

### 2.2. Statistical Indicators of Accuracy in Electric Power Forecasting

The number of papers from the publications we studied that are eligible according to the criteria was 164. We collected the mean absolute percentage error (MAPE), a statistical indicator of accuracy. *This index indicates an average of the absolute percentage errors (Equation (Equation 1)); the lower the MAPE, the higher is the accuracy* [27].
(1)$$MAPE=\frac{1}{m_{k}}\sum_{k=1}^{m}\left|\frac{t_{k}-y_{k}}{t_{k}}\right|*100$$
where $t_{k}$ is the actual value of electric power, $y_{k}$ is the forecasting value produced by the model, and *m* is the total number of observations. The final quality-controlled database from the 164 documents contained 4883 entries (MAPE, type of MAPE, country, date, input variables, model, type of model, latitude, longitude, and size of sample), and we saved 257 entries associated with a MAPE value linked to the best model proposed in the cases of study with data from 33 countries. The locations are represented on the world map in Figure 4. We can see that the studied publications cover all continents. Occupying the first positions in the list of the countries with the most electrical energy forecast documents, under the criteria used, are Australia, China, Iran, and Turkey.

## 3. Description of the Dataset

The analysis was performed from five perspectives: the class of a forecasting model (MSC or ML), the type of model (hybrid or not), the time horizon, and the input variables and performance trend over time (MAPE). The dataset analyzed in this paper contains 257 entries associated with a MAPE value linked to the best model proposed in the document.

The MAPE value was classified according to the criteria drawn up by [28], which contain typical MAPE values for business and industrial data and their interpretation in four evaluation criteria (in our case, four prediction capabilities); this table was used in [29,30,31,32], and can be seen in Table 1.

Table 2 shows that of the 164 documents processed in the systematic review, 99 contain a highly accurate prediction (HAP). Additionally, more ML documents with an HAP were found than MSC documents. Regarding the sources of variability considered by the documents that contained HAPs, it can be seen that multivariate models have a higher recurrence than univariate models. As explained in [10], despite the introduction of artificial intelligence, each of the individual methods are still not able to produce the desired outcomes because of their disadvantages. For instance, neural networks attain local optimal results instead of global optimal results. Expert systems excessively rely on knowledge and cannot always obtain optimal results, whereas grey prediction systems are suitable for exponential growth models. Thus, by considering every method’s merits and taking full advantage of them, the concept of hybrid and combination methods developed rapidly.

### 3.1. Forecasting Horizon

Figure 5 shows that the minimum MAPE values (<2) were reached more frequently when the forecast time horizon is 5 min.

### 3.2. Exogenous Influence

Because forecasting electric power demand is typically based on historical electricity consumption and its relationship with exogenous influences, such as gross domestic product (GDP), population, urbanization, income and exports, research on forecasting electric power demand has evolved using both univariate and multivariate time-series models [15].

Similarly, weather associated variables such as humidity, temperature, and dew point are pertinent for electric power forecasting for extensive time scales. For short-term forecasting such as minutes ahead, the climate changes are already captured in the electric power series [165]. Forecasting models using only previous electricity data (univariate) have been shown to provide HAP and to perform better than models that also use weather variables as exogenous influences (multivariate) [166]. Nevertheless, the use of weather influences was found to be beneficial for electric power forecasting horizons beyond several hours [166,167]. In Figure 6, it can be seen that the precision of the electric energy forecast on average tends to improve when various sources of variability are considered. In our sample of filtered documents, the average MAPE is lower for the forecasts whose models consider sources of variability from the calendar information, weather information, and economic or sociodemographic information, as in [60,125,130,146].

## 4. Classes of Forecasting Models

From the multitude of methods that have been tested and evaluated, the ML and MSC classes seem to be the main competitors.

### 4.1. Classical Statistical Models

A popular technique such as time series forecasting is applied in several areas [25]. The most widely used statistical method is the ARIMA of Box and Jenkins, which was applied with more force during the eighties, when intelligent systems began to appear [168]. Several time series models make use of the high autocorrelation for small lags in the time series of electric power, and supply electric power forecasts using only previously measured values of electric power as input.

From the multitude of methods that have been tested and evaluated in this review, in this class, regression analysis and ARIMA modeling seem to be the main competitors (Figure 7).

### 4.2. Classical Regression in the Time Series Context

To explain linear regression in the the context of time series, we assume some output or dependent time series. Assume $x_{t}$ for t=1,⋯,n, is being influenced by a collection of possible inputs or independent series, such as $z_{t_{1}},z_{t_{2}},\cdots ,z_{t_{q}}$, where we first regard the inputs as fixed and known [169]. We express this relation through the linear regression model:(2)$$x_{t}=\beta_{0}+\beta_{1}z_{t1}+\beta_{2}z_{t2}+\cdots +\beta_{q}z_{tq}+w_{t}$$
where $\beta_{0},\beta_{1},\cdots ,\beta_{q}$ are unknown fixed regression coefficients, and $w_{t}$ is a random error or noise process consisting of independent and identically distributed (iid) normal variables with a zero mean and variance $\sigma_{w}^{2}$. For time series regression, it is rarely the case that the noise is white, and we will need to eventually relax that assumption.

Classical regression models have been used in several academic papers for electric power forecasting [97,98,102,124,130,134,138,140], reaching an accuracy in the forecast with an average MAPE value of 1.569%. Classical regression is often insufficient for explaining all of the interesting dynamics of a time series; instead, the introduction of correlations that may be generated through lagged linear relations led to the autoregressive (AR) and autoregressive moving average (ARMA) models that were presented in [169,170]. Adding nonstationary models to the mix led to the autoregressive integrated moving average (ARIMA) model popularized in the landmark work by Box and Jenkins [169,171].

### 4.3. Autoregressive Integrated Moving Average

Autoregressive models are based on the idea that the current value of the series, $x_{t}$, can be explained as a function of *p* past values, $x_{t-1},x_{t-2},\cdots ,x_{t-p}$, where *p* determines the number of previous steps required to forecast the current value [169].

The acronym ARIMA refers to an autoregressive integrated moving average model. ARIMA models can be applied to non-stationary data, and when the data are seasonal, the SARIMA model can be implemented. The ARIMA and SARIMA models have been used in many studies for forecasting [14,15,16,99,100,103,127,136], reaching forecast accuracies with an average MAPE value of 3.214%. A typical ARIMA (p,d,q) model can be expressed by Equation (Equation 3), where the variable $u_{t}$ is replaced by a new variable $w_{t}$ obtained by differencing $u_{t}$
*d* times [25]:(3)$$w_{t}={\left(1-B\right)}^{d}u_{t}.$$

### 4.4. Machine Learning (ML) Models

ML methods have been suggested in the academic literature as an alternative to MSC methods for time series prediction, with the same objective. They attempt to improve the forecasting accuracy precision by minimizing some loss functions, as for example the sum of squared errors. The distinction between ML and MSC is in how the minimization is performed: the ML methods use nonlinear algorithms while the MSC method use linear processes. The ML methods require a greater dependence on computer science to be implemented and are more demanding than MSC methods, as they are positioned at the intersection of MSC and computer science [172]. There are several approaches developed under ML theory. In this review, artificial neural networks (ANNs), support vector machines (SVMs), decision trees (DTs), adaptive neuro fuzzy inference systems (ANFISs), and recurrent neural networks (RNNs) were found to support the bases of the models that were implemented more frequently in electric power forecasting (Figure 7).

#### 4.4.1. Artificial Neural Networks (ANN)

Neural networks have been the subject of great interest for many decades due to the desire to understand the brain and to build learning machines [173]. *A neural network is an interconnected assembly of simple processing elements, units or nodes whose functionalities are loosely based on animal neurons. The processing ability of a network is stored in the inter-unit connection strengths, or weights, obtained by a process of adaptation to, or learning from, a set of training patterns* [174].

The ANN models have been used in many studies for electric power forecasting [6,39,44,46,48,53,54,55,57,58,59,60,61,62,65,70,75,77,81,82,86,108,153,155,165,175,176] and have reached a forcasting accuracy with an average MAPE value of 3.781%.

#### 4.4.2. Recurrent Neural Networks (RNN)

Models known as a recurrent neural networks allow feedback connections; these models define nonlinear dynamical systems but do not have simple probabilistic interpretations [173]. RNN models have been used in many studies for electric power forecasting [64,69,71,73,88,90,157,177,178] and have reached a forecasting accuracy with an average MAPE value of 3.610%.

#### 4.4.3. Fuzzy Neural Network-Based Forecasting Methods

Fuzzy logic systems (or, simply, fuzzy systems (FSs)) and neural networks are universal approximators; that is, they can approximate any nonlinear function (mapping) with any desired accuracy and have found wide application in the identification, planning, and model-free control of complex nonlinear systems, such as robotic systems and industrial processes. Fuzzy logic offers a linguistic (approximate) approach to drawing conclusions from uncertain data, and neural networks offer the capability of learning and training with or without a teacher (supervisor) [179].

Fuzzy logic algorithms have been used in many studies for electric power forecasting [10,27,33,45,50,52,63,79,80,83,85,91,113,118,122,154,160,180,181] and have reached a forecasting accuracy with an average MAPE value of 4.013%.

#### 4.4.4. Support Vector Machines (SVMs)

Support vector machines are supervised learning algorithms used for solving binary classification and regression problems. The main idea of support vector machines is to construct a hyperplane such that the margin of separation between the two classes is maximized. In this algorithm, each of the data points is plotted as a data point in n-dimensional hyperspace. Then, a hyperplane that maximizes the separation between the two classes is constructed [182]. *This technique was originally designed for binary classification but can be extended to regression and multiclass classification* [173]. Support vector regression algorithms have been used in many studies for electric power forecasting [41,47,49,76,109,110,112,114,119,164,183,184,185] and have reached a forecasting accuracy with an average MAPE value of 4.326%.

## 5. Evaluation of Model Accuracy

As can be seen in Table 3, as mentioned in [77], there are many factors, such as economic development, regional industrial production, holiday periods, weather conditions, social change, electricity price, and population, that are unavoidable, affect electric power randomly, and allow the data to demonstrate different features.

Short-term load forecast models that rely on weather information require the prediction of weather parameters for the next few hours or at most the next few days [75]. Similarly, economic indicators and electrical infrastructure measures are usually useful in forecasting electric power with a long forecast horizon, e.g., a prediction of the annual peak load at least one year in advanced [39]. However, in the daily peak load forecasting for the following month, these indicators are not effective, since the forecast step and horizon are too short to observe their effect [75]; this behavior is shown in Table 3.

Similarly, from Figure 8, it can be seen that among the records of documents that reach HAP, the average MAPE value is lower in the frameworks that implement hybrid models of ML and multivariate dependency, such as those developed in [6,27,73,74,75,76,77,78,79,80,81,82,83,84,85,86,87,88,89,90,91]. To verify the hypotheses of the differences in the means and variances in the MAPE, three hypothesis tests are carried out. Table 4 shows that for small and medium effects, the alternative hypothesis on the minor indicates that the MAPE is accepted for the ML model, based on a hybrid method and with multivariate dependencies.

In this sense, a summary of the documents found in the review with MAPEs and HAPs that base their models on ML with a hybrid approach and multivariate dependence is presented in the Table 3. When we analyzed these documents, we observed that there are common elements; for example, when building a word cloud from the abstracts, keywords and titles of these documents, we can identify that in 24% of the cases multiple scale decomposition and wavelet theory (WT) were mentioned.

The wavelet transform, including filtering and forecasting, has been suggest for detailed examination of the elements or structure of time series in several academic papers in recent years [73]. WT has been extensively implemented in electric power forecasting for decomposing electricity series into series with particular characteristics that can be predicted more accurately than the original time series [186,187,188].

## 6. Case Study

In this section, we propose a hybrid model to forecast the electric power by using a type of recurrent artificial neural network known as long short-term memory (LSTM), developed by [189]; we also implemented wavelet decomposition for the data preprocessing (WD-LSTM), as was used in [90]. We use the acronym WD-LSTM for the proposed hybrid model. The results were compared with those of traditional neural network models (LSTM), as was applied in [71,177] and with results of the lagged regression analysis as in [96].

The performance of this methods is demonstrated with a case study using an actual dataset collected from Chile (Table 5). The objective is to illustrated the approach that allows the electric power demand forecasting, in terms of its lagged values, identifying the type of model that tends to show better forecast accuracy.

Maximum daily and hourly electric power demand data over a diverse period were used (Table 5). Figure 9 shows that there is a regularity in electric power demand data. We observe a clear pattern based on the year and day of the week. The electric power demand also follows a group of patterns within any day and depending on the time of the day.

The values of four performance evaluation indicators—RMSE: root mean square error, MAE: mean absolute error, $R^{2}$: coefficient of determination, and MAPE: mean absolute percentage error—showed that the hybrid deep learning model (WD-LSTM) exhibits superior performance in both forecasting accuracy and stability.

Figure 10 and Figure 11 provide the comparative hourly and daily-ahead performance results for three types of days (weekday, weekend, and all days). They likewise provide the performance evaluation results of regression, LSTM, and WD-LSTM applied to each dataset (training, validation, and test). The hybrid deep learning model (WD-LSTM) had the best performance of all forecasting models. The WD-LSTM method generated forecasting results with the lowest MAE, MAPE, and RMSE and with the higher $R^{2}$ in most cases. The results further reveal the robustness of the hybrid deep learning model. The superior accuracy of the hybrid model is primarily due to the deep learning framework comprising between two and four independent LSTM networks, which provide an effective means to approximate inherent invariant features and structures. In addition, the low- and high-frequency components exhibited in the electric power datasets can be better extracted by wavelet decomposition. Likewise, each LSTM network managed to focus more on capturing the linear and nonlinear relationships in the energy series, which could not be done with the lagged regression, at least in non-linear cases.

## 7. Discussion and Conclusions

This paper presented a systematic review of the forecasting models for electric power from the last 15 years based on ML and MSC techniques. We presented an in-depth analysis of the performance of electric power forecasting models and compared different forecasting models based on their MAPE values. A rigorous framework for comparing different classes of models was introduced, thus generating a reliable picture of the state-of-the-art models’ accuracy of electric power forecasting. We were able to identify that a large number of techniques are being used and are aimed at forecasting electrical energy; the techniques with the greatest use are in the fields of ML and ANNs, followed by those that implement algorithms with fuzzy logic and RNNs, while in the MSC area, the use of ARIMA models and regression analysis predominates.

The results can be stratified from three perspectives. The forecasting models (I) from the hybrid class, (II) of multivariate dependency, and (III) based on the machine learning approach demonstrate the best performance for electric power forecasting. Regarding the hybrid models, it is highlighted that 24% of the adjustments with the greatest forecasting precision merged wavelet theory into their models. With regard to multivariate models, we were able to identify that those models that incorporate various sources of variability in their adjustment tend to have, on average, greater precision in their forecasts.

A case of study was presented, in which the implementation of MSC and ML models was compared; we found that the linear models, such as lagged regression, are relatively simple and cannot capture with precision the inherent nonlinear structure of the electric power time series, whereas the deep learning models implemented have a better performance.

Likewise, it was observed that when decomposing the series according to the type of day of electricity consumption (workday or weekend), the models tend to have better forecast accuracy and, in the same way, forecasting errors are minimized by reducing the time horizon (hourly).

Due to electric power systems’ participation in the growing trend of environmental optimization around the world, a substantial increase in the contribution of diverse sources to the energy generation is observed. This trend brings about challenges in terms of electric power generation and distribution system operation, because the dimension and complexity of such advances, among other aspects, require the use of a computational intelligence systems that act as sources of data and deal with the control, management, and trading needs at the distribution level in an efficient and robust manner. In this sense, further research could deepen the understanding of the relationship between the type of energy, climate, preprocessing techniques, and performance of machine learning models under various normalized metrics of residuals.

## Figures and Tables

**Figure 1 entropy-22-01412-f001:**
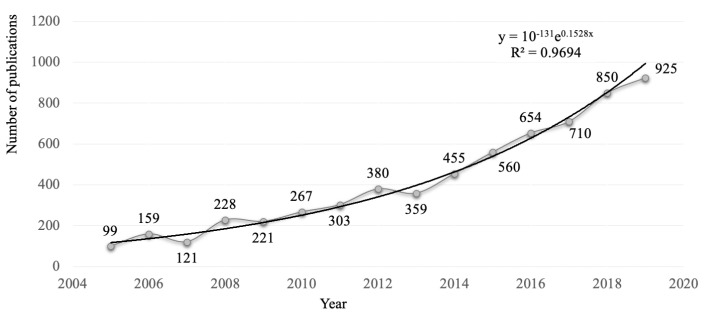
Number of articles published per year. The line represents an exponential fit, highlighting the yearly growth trend. The publications from 2020 are excluded since only partial data are available.

**Figure 2 entropy-22-01412-f002:**
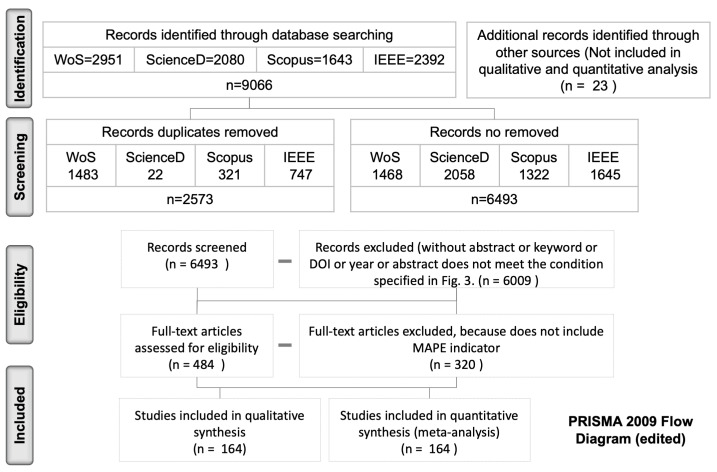
PRISMA Flow Diagram.

**Figure 3 entropy-22-01412-f003:**
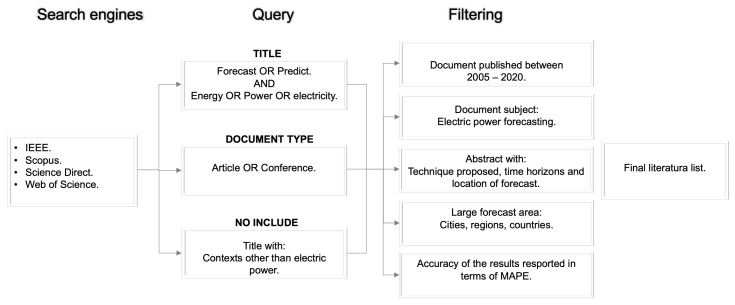
Search methodology for finding relevant literature.

**Figure 4 entropy-22-01412-f004:**
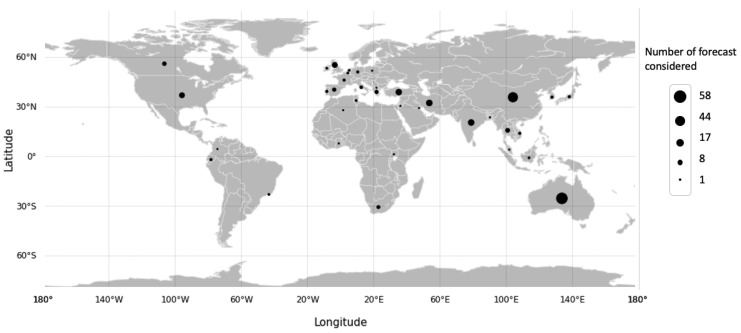
Number of forecasts by country considered in the review.

**Figure 5 entropy-22-01412-f005:**
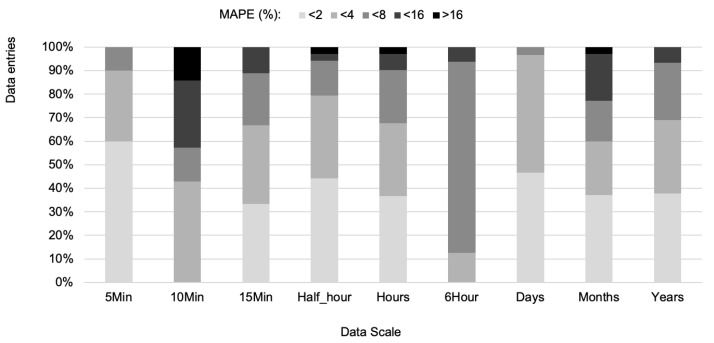
MAPE value interval and the percent of forecasts extracted from the 164 documents considered by time horizon.

**Figure 6 entropy-22-01412-f006:**
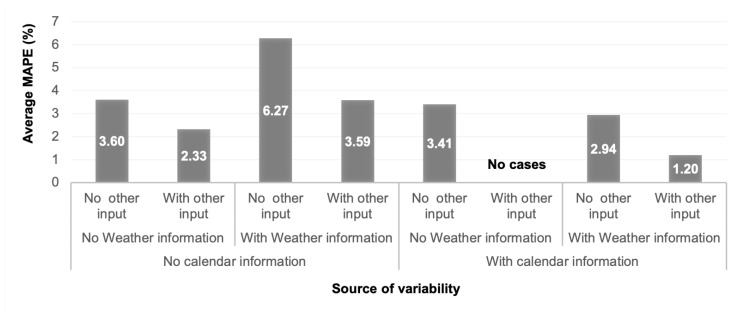
Average MAPE according to the considered source of variability.

**Figure 7 entropy-22-01412-f007:**
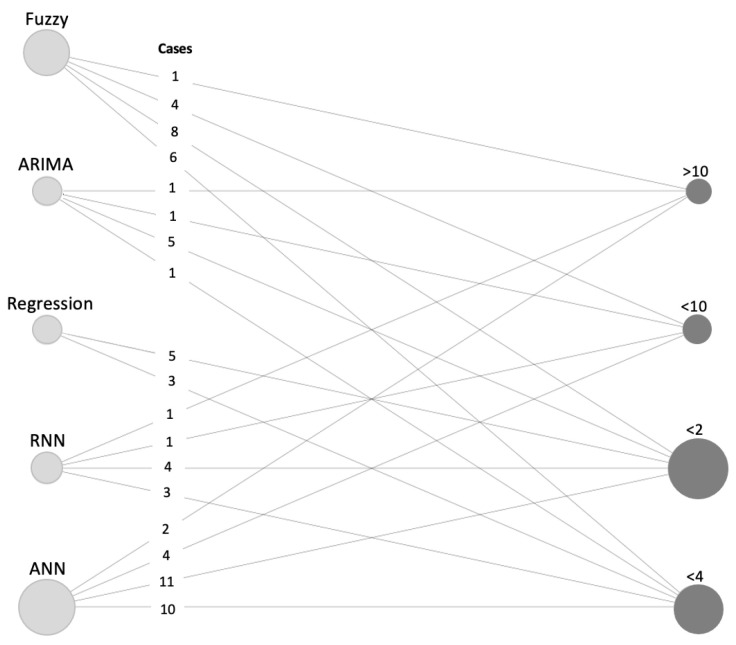
Graph with nodes weighted by sample size for the case of the techniques (light-gray node) and probability of occurrence for the MAPE intervals (dark-gray node).

**Figure 8 entropy-22-01412-f008:**
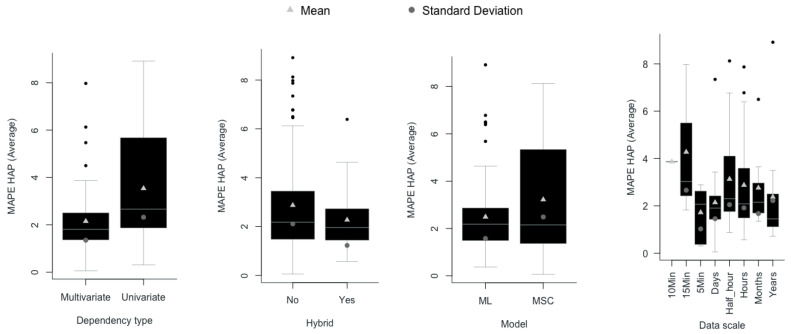
Boxplot of the papers included in the systematic review, with HAP-MAPE.

**Figure 9 entropy-22-01412-f009:**
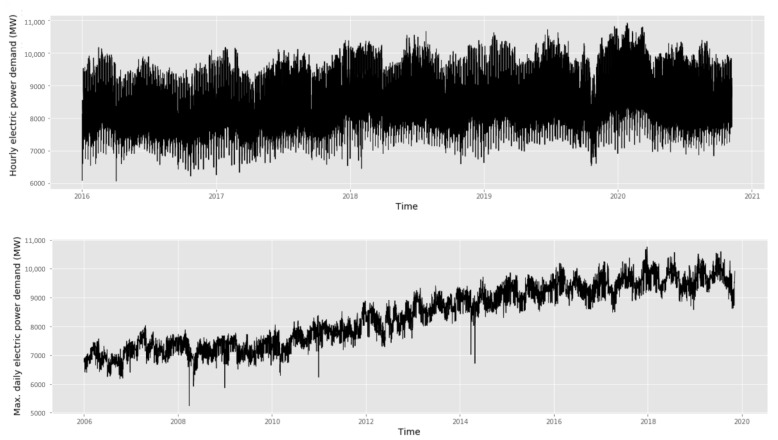
Electric power demand series (MW) in Chile.

**Figure 10 entropy-22-01412-f010:**
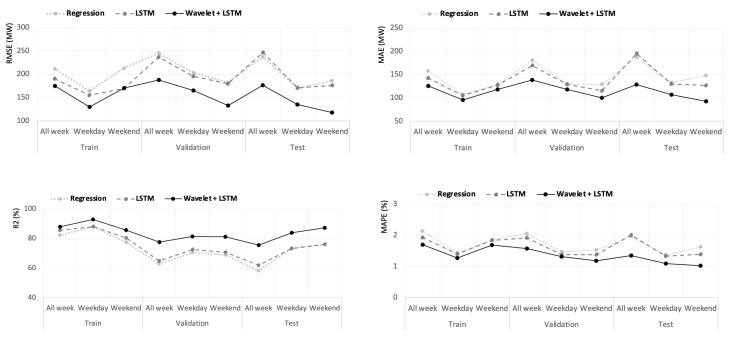
Performance evaluations of different methods for each type of day.

**Figure 11 entropy-22-01412-f011:**
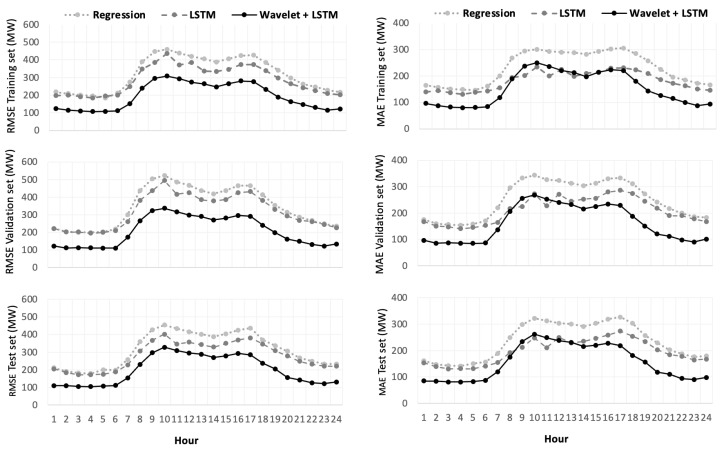
Performance evaluations of different methods for hour.

**Table 1 entropy-22-01412-t001:** MAPE qualitative criteria.

MAPE (%)	Prediction Capability
<10	Highly accurate prediction (HAP)
10–20	Good prediction (GPR)
20–50	Reasonable prediction (RP)
>50	Inaccurate prediction (IPR)

**Table 2 entropy-22-01412-t002:** Systematic review documents. Techniques type used in electric power forecast and qualitative values of the average MAPE.

MAPE	Type	Multivariate Model		Univariate Model		Total
		Not Hybrid	Hybrid	Not Hybrid	Hybrid	
HAP	ML	[33,34,35,36,37,38,39,40,41,42,43,44,45,46,47,48,49,50,51,52,53,54,55,56,57,58,59,60,61,62,63,64,65,66,67,68,69,70,71,72]	[6,27,73,74,75,76,77,78,79,80,81,82,83,84,85,86,87,88,89,90,91]	[1,14,17,18,92,93,94,95,96,97,98,99,100,101,102,103,104,105,106,107]	[29,32,85,108,109,110,111,112,113,114,115,116,117,118,119,120,121,122]	99
HAP	MSC	[15,123,124,125,126,127,128,129,130,131,132,133,134,135,136,137,138,139,140,141,142,143,144,145,146,147]	[148,149]	[1,14,17,18,92,93,94,95,96,97,98,99,100,101,102,103,104,105,106,107]	[19,150,151,152]	52
GPR	ML	[153,154,155,156,157]	[158]	[159,160,161,162]	−	10
GPR	MSC	[16,163]	−	−	−	2
RP	ML	[164]	−	−	−	1
**Total**		**74**	**24**	**44**	**22**	**164**

**Table 3 entropy-22-01412-t003:** Summary of the papers included in the systematic review, with a HAP-MAPE, hybrid ML forecasting approach and multivariate model. The abbreviations are displayed in Abbreviations Section. ${}^{1/}$ Average MAPE; ${}^{2/}$ Approximate sample size. The Classification of the forecasting models can be seen in Figure A1.

Ref	Year	Country	Energy Type	Technique	Forecast	Other Input	MAPE ${}^{1/}$	N ${}^{2/}$	Scale	Date Sample	
**[73]**	2006	Australia	No Specific	ERNN; WT	Electricity Load	TM, HM, WS	0.794	26,297	Hours	1999	2002
**[74]**	2013	Iran	Wind	PSO; ACO	Wind Power	TM, WS	3.513	8736	Hours	2010	2011
**[75]**	2008	Iran	No Specific	ANN; EA	Peak Load	CI	1.760	26,280	Hours	1997	1999
**[76]**	2008	EEUU	No Specific	SVR; BT	Electricity Load	CI, TM, HM	1.960	30,144	Hours	2001	2004
**[77]**	2015	Australia	No specific	ANN	Electrical power	CI	3.710	70,080	Half-Hour	2006	2009
**[78]**	2017	UK	No Specific	PSO; ANN	Load Demand	CI, TM	1.723	8760	Hours	2008	2008
**[79]**	2016	Algeria	No Specific	HW-ES; KNN; WD; Fuzzy-CM; ANFIS	Peak Electricity	TM	2.796	1064	Days	2012	2014
**[80]**	2010	Iran	No Specific	ANFIS	Electricity	GDP, POP, EXP, CPI	2.789	37	Years	1971	2007
**[81]**	2017	Poland	No Specific	ANN; PCA	Power Load	CI, TM	1.235	26,280	Hours	2009	2012
**[82]**	2018	India	No Specific	ANN; PSO; GA	Electricity Demand	CPI, GDP	0.220	25	Years	1991	2015
**[83]**	2017	UK	No Specific	ELM; Fuzzy	Electricity Load	CI, TM, DP	1.435	43,852	Hours	2004	2008
**[84]**	2019	EEUU	Wind	NWP; WD; CNN	Wind Power	CI, TM, WS, DP	2.550	26,280	Hours	2015	2017
**[85]**	2008	Iran	No Specific	BNN; MCM; Fuzzy	Load	CI, TM	2.421	1460	Days	2004	2007
**[27]**	2018	Vietnam	No Specific	WT; ANFIS; COA	Electricity	CI, TM, HM, PRS, RFL, RT, WS	4.330	132	Months	2003	2013
**[86]**	2017	UK	No Specific	ANN; JOA	Electricity Load	CI, TM, DP	5.710	52,560	Hours	2004	2009
**[87]**	2015	India	No Specific	ANN; BBO	Electrical Energy	GDP, POP	2.510	33	Years	1980	2012
**[88]**	2019	China	Wind	GM; ERNN; BP	Power Generation	TM, HM, WS, WDD, PRS	3.730	1441	15 min	2016	2016
**[6]**	2019	Australia	No Specific	ANN; BOOT	Electricity	57 Index	5.290	4300	6 h	2014	2017
**[89]**	2019	Uganda	No Specific	PSO; ABC	Electricity	POP, GDP, EP, NS	1.306	17	Years	1990	2016
**[90]**	2020	Australia	Photovoltaic	WD; LSTM	Power	TM, HM, WS, HR	1.868	213,984	5 min	2014	2016
**[91]**	2018	Turkey	No Specific	ANFIS	Electrical Load	CI, TM	8.869	8760	Hours	2017	2017

**Table 4 entropy-22-01412-t004:** Hypothesis Test for Difference in Means. ${}^{1/}$ Levene Test (*p*-value); ${}^{2/}$ test (*p*-value).

Variable	Hypothesis $\left(H_{0}\right)$	Homogeneity of Variance ${}^{1/}$	Difference in Means ${}^{2/}$	Effect Size (Cohen’s)
Model	$\mu_{{}_{ML}}\geq \mu_{{}_{MSC}}$	0.00386	0.07252	Small
Hybrid	$\mu_{{}_{Yes}}\geq \mu_{{}_{No}}$	0.09063	0.04321	Small
Dependency	$\mu_{{}_{Multi}}\geq \mu_{{}_{Uni}}$	0.00125	0.00059	Medium

**Table 5 entropy-22-01412-t005:** Electric power demand object of forecasting.

Type	Variable		Date	Set Size		
				Training	Validation	Test
**Local** **Energy**	Dmax	Maximum daily electricity demand (MW).	2006–2019	2475	1516	1062
	Hed	Hourly electricity demand (MW).	2016–2020	865	371	530

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
