# Peer review of "A Systematic Review of Statistical and Machine Learning Methods for Electrical Power Forecasting with Reported MAPE Score"

_entropy, 2020, doi:10.3390/e22121412_

Round 1

Reviewer 1 Report

In this paper, a systematic literature review is presented to identify the type of model with the highest tendency to show precision in the context of electric power forecasting. Two classes of forecasting models were compared: classical statistical or mathematical and machine learning models. The use of hybrid models is identified, which have made significant contributions to electric power forecasting. The review work presented is timely and the article is well written however can be improved further based on the following suggestions/comments.

1. The contributions and novelty of the work are too little. 

2. The article is mostly focusing on an overview of papers published on forecasting rather than more detailed investigation on the presented models and their individual finding.  A case study (not error comparison only) can be presented based on these models and compare based on these models and presented the critical findings which can help future researchers to choose the most appropriate model for their research. 

3. The findings of the proposed work is comprehensive, not specific and critical. 

4. A case study on renewable or load forecasting would increase the significance of the content.

5. Very light text colour is used in most of the figure reduces visibility and

appearance. 

Author Response

Response to Reviewer 1:

Thank you for your review of our paper. We have answered each of your points below

Reviewer’s comment 1:  the contributions and novelty of the work are too little. 

Response:  We appreciate this comment, certainly, it is an active area of research, so it is important to have some insight to know where a researcher can contribute with a novel research.  Nevertheless, we have incorporated a case study that supports the descriptions obtained in the metrics associated with the performance of the models, with the purpose to provide a framework of substantial guidance to researchers in the area around the techniques that tend to promote precision in the forecast, highlighting the benefit shown by the implementation of deep learning for forecasting in the electric power demand and bearing in mind that the hybridization of processes tends to optimize the results, which is why it is the path that will probably be applied more frequently in the future. (p. 12, line 248).

Reviewer’s comment 2: the article is mostly focusing on an overview of papers published on forecasting rather than more detailed investigation on the presented models and their individual finding.  A case study (not error comparison only) can be presented based on these models and compare based on these models and presented the critical findings which can help future researchers to choose the most appropriate model for their research. 

Response: We agree with the reviewer regarding this comment. This work is a systematic review that evidences the results that have been obtained in the literature in relation to the prediction of electrical energy, however we have added our own hybrid model with a case study using an actual dataset collected from Chile. (p. 12, line 248).

Reviewer’s comment 3: the findings of the proposed work is comprehensive, not specific and critical. 

Response: We added several paragraphs in the discussion section that addresses this comment (p. 15, line 296-311).

Reviewer’s comment 4: a case study on renewable or load forecasting would increase the significance of the content.

Response: We agree with the reviewer and have added a complete section called "Case of Study" (p. 12, line 248).

Reviewer’s comment 5: very light text colour is used in most of the figure reduces visibility and appearance. 

Response:  We appreciate this assessment, We have made the suggested changes (p. 9, line 219, p. 10, line 247)

We appreciate your comments and we hope you like the modifications made, we are attentive to any criticism or recommendation that you may submit to us.

Reviewer 2 Report

A literature review of electrical power forecasting is proposed. After describing the strategy for considering an article for this review, a descriptive analysis of the prediction performances is given.

The idea of this work is interesting since a massive production of articles on this area arrived in the last 20 years. Different type of electric power forecasting exists, aiming at different time and space resolutions. All the references are mixed together without a clear classification among these different problems.

The task is difficult, and unfortunately some decisions taken to filter documents narrower the reach of the conclusions of the work. Indeed, the techniques that are the state-of-the-art in the domain are missing in the references. Moreover, the important problems today in the domain are completely absent of the work.

The work is oversold, with conclusions that are too tightly connected to the filtering choices of the authors.

Author Response

Response to Reviewer 2:

Thank you for your review of our paper. We have answered each of your points below

Reviewer’s comment 1: the task is difficult, and unfortunately some decisions taken to filter documents narrower the reach of the conclusions of the work. Indeed, the techniques that are the state-of-the-art in the domain are missing in the references. Moreover, the important problems today in the domain are completely absent of the work. 

We answer after the next comment.

Reviewer’s comment 2: The work is oversold, with conclusions that are too tightly connected to the filtering choices of the authors.

Response:  We appreciate this comment, we have incorporated a section trying to explain with greater precision the implementation of the PRISMA methodology as a basis to develop the review (p3, line 68). 

Also, due to the participation of electric power systems in the  growing trend of environmental optimization in the world, a substantial increase in the contribution of diverse sources to the energy generation is observed. This trend is accompanied by challenges in terms of the electric power generation and distribution system operation, because the dimension and complexity of such advances that, among other aspects, require the accompaniment of a computational intelligence that act as sources of data, to deal with the control, management, and trading needs at the distribution level in an efficient and robust manner. 

The various sources of energy generation observed require specific treatments as they respond to diverse sources of variability and on multiple scales, however we hope to contribute with the incorporation of a complete section called "Case of Study" and with the added of diverse paragraphs in the discussion section that does the most critical work (p12, line 248, p. 15, line 296).

Addressing the specificities of the ideal forecasting method for each possible source of energy generation becomes uphill for a single document, however we try to illustrate with the case study the benefits of the implementation of deep learning techniques, which have great acceptance in recent investigations framed in the context of the forecast of electrical energy and for various sources of generation of the same, as observed in table 4.

We appreciate your comments and we hope you like the modifications made, we are attentive to any criticism or recommendation that you may submit to us.

Round 2

Reviewer 1 Report

No more comments, Congratulations!

Author Response

Thank you!

Reviewer 2 Report

The authors add details about their screening procedure for selecting articles. It seems to be based on a set of more or less automatic rules. My remarks stated for the first version hold, the procedure filters out too many important references and mixes up works coming from different problems (supply/demand forecasting, different prediction horizons, different time resolutions, etc).

I am not able to evaluate the screening procedure itself (which seems not to be the focus of the article). However, I maintain that its application on the electrical demand forecast (EDF) gives very poor results. No important state-of-the-art work is comprised. The conclusions are overstated since the authors only view the part of the bibliography that was selected by their automatic system.

The new case study is interesting, although quite disconnected from the rest of the article.

Author Response

Word adjunt
